# The Efficacy of Short-Term Weight Loss Programs and Consumption of Natural Probiotic Bryndza Cheese on Gut Microbiota Composition in Women

**DOI:** 10.3390/nu13061753

**Published:** 2021-05-21

**Authors:** Ivan Hric, Simona Ugrayová, Adela Penesová, Žofia Rádiková, Libuša Kubáňová, Sára Šardzíková, Eva Baranovičová, Ľuboš Klučár, Gábor Beke, Marian Grendar, Martin Kolisek, Katarína Šoltys, Viktor Bielik

**Affiliations:** 1Department of Biological and Medical Science, Faculty of Physical Education and Sport, Comenius University in Bratislava, 814 69 Bratislava, Slovakia; ivan.hric@uniba.sk (I.H.); simona.ugrayova@uniba.sk (S.U.); libusa.kubanova@uniba.sk (L.K.); 2Biomedical Center, Institute of Clinical and Translational Research, Slovak Academy of Sciences, 845 05 Bratislava, Slovakia; adela.penesova@savba.sk (A.P.); zofia.radikova@savba.sk (Ž.R.); 3Department of Microbiology and Virology, Faculty of Natural Sciences, Comenius University in Bratislava, 842 15 Bratislava, Slovakia; sardzikova2@uniba.sk (S.Š.); katarina.soltys@uniba.sk (K.Š.); 4Biomedical Center Martin, Jessenius Faculty of Medicine in Martin, Comenius University in Bratislava, 036 01 Martin, Slovakia; eva.baranovicova@uniba.sk (E.B.); marian.grendar@uniba.sk (M.G.); martin.kolisek@uniba.sk (M.K.); 5Institute of Molecular Biology, Slovak Academy of Sciences, 845 51 Bratislava, Slovakia; lubos.klucar@savba.sk (Ľ.K.); gabor.beke@savba.sk (G.B.); 6Comenius University Science Park, Comenius University in Bratislava, 841 04 Bratislava, Slovakia

**Keywords:** weight loss program, bryndza cheese, body composition, gut microbiota

## Abstract

Weight loss interventions with probiotics have favourable effects on gut microbiota composition and derived metabolites. However, little is known about whether the consumption of natural probiotics, such as Bryndza cheeses, brings similar benefits. The purpose of the study was to find the effect of short-term weight loss programs and Bryndza cheese consumption on the structure of the gut microbiota, microbiota-derived metabolites and body composition in middle-aged women. We conducted a randomised controlled intervention study. Twenty-two female participants with a body fat percentage ≥25% underwent a short weight loss program (4 weeks). Subjects were randomised to either the control or intervention group according to diet. The intervention group comprised 13 participants, whose diet contained 30 g of “Bryndza” cheese daily (WLPB). The control group comprised nine participants without the regular consumption of Bryndza cheese (WLP) in their diet. Both interventions lead to a significant and favourable change of BMI, body fat, waist circumference and muscle mass. Moreover, the relative abundance of *Erysipelotrichales* significantly increased in both groups. However, the relative abundance of lactic acid bacteria (*Lactobacillales*, *Streptococcaceae*, *Lactococcus* and *Streptococcus*) significantly increased only in the WLPB group. Furthermore, short-chain fatty acid producers *Phascolarctobacterium* and *Butyricimonas* increased significantly in the WLPB group. A short-term weight loss program combined with Bryndza cheese consumption improves body composition and increases the abundance of lactic acid bacteria and short-chain fatty acid producers in middle-aged women.

## 1. Introduction

There is an abundance of literature on the positive effects of active lifestyle programs, diet, sport and exercise training on weight loss, metabolic health and prevention of chronic noncommunicable diseases [1,2,3,4]. However, it is difficult to find the optimal strategy; basically, weight loss programs differ in terms of diet, calorie intake, type, intensity and frequency of physical activity (if included).

Several studies have revealed that interventions including diet and moderate physical activity have a greater joint effect on body weight and composition than interventions featuring just one of these components [5,6]. Low caloric diet without physical activity is effective for short-term weight loss, but physical activity is vital for modifying body composition and improving physical fitness [7]. In addition, losing weight through an active lifestyle may modulate gut microbiota with a positive impact on metabolic regulation [8,9].

Active lifestyle programs are often associated with negative energy balance, diet composition or preferring some functional foods consumption [10,11,12,13]. Food containing probiotics is a functional food with a relative long history. Probiotics are defined by the World Health Organization (WHO) as “live microorganisms which when administered in adequate amounts, confer a beneficial health effect on the host” [14]. Traditional and efficient probiotics products are fermented dairy products [15]. Yogurt is one of the best-known fermented dairy foods [16,17] that has positive health effects [18,19]. Nevertheless, there are other natural fermented foods containing several probiotic strains [20]. Fermented sheep cheese from pasteurised milk (Bryndza) is a traditional dairy product from Slovakia [21]. Bryndza was recently granted the protected geographic indication (PGI), as it is produced in the defined Carpathian Mountains region of Slovakia [22,23]. Slovaks have been eating Bryndza cheese presumably since the 14th century. However, the word “Bryndza” comes from the Wallachian word, which Romanians used to name any type of salted cheese [24]. It is a natural, spreadable, white food made of fermented ripened ewe’s cheese [25]. Recent studies showed that the majority of Bryndza’s microbiota is formed by lactic acid bacteria such as *Lactococcus*, *Streptococcus*, *Lactobacillus* and *Enterococcus* [26,27,28]. The positive effects of lactic acid bacteria (LAB) on the structure and modulation of gut microbiota and metabolism are well known [29,30,31]. However, there is scarce evidence on the influence of regular Bryndza cheese consumption on gut microbiome and blood metabolites [32]. 

The purpose of this study was to find the effect of a short-term weight loss program on body composition and the structure of gut microbiota and selected blood metabolites. At the same time, we aimed at elucidating benefits of the consumption of Bryndza cheese on microbiome and metabolic variables.

## 2. Materials and Methods

### 2.1. Subjects

A total of 30 participants agreed to participate in the study, of which 22 met the inclusion criteria. The cohort included healthy females who signed up for a short-term weight loss program. The participants were allocated into 2 groups: (a) healthy females (average age 44 ± 13.0 years) who completed a 30-day weight loss program (WLP), and (b) healthy females (average age 51 ± 12.6 years) who completed a 30-day weight loss program that included the regular consumption of natural probiotics (Bryndza cheese). 

Inclusive criteria were: (1) females aged 18–65 years and (2) BMI 20–40 kg/m^2^. Exclusive criteria included: (1) the use of antibiotics for a period of 2 weeks (wk) prior to the study, (2) supplementing probiotics 2 months prior to the study, (3) acute or chronic diseases or infections (including upper respiratory tract infections, fever, chronic inflammatory disorders, autoimmune disorders) 2 months prior to the study, (4), alcohol consumption, (5) smoking or drug abuse, (6) history of digestive diseases (such as inflammatory bowel disease, irritable bowel syndrome) and (7) previous gastrointestinal surgery. 

### 2.2. Intervention

#### 2.2.1. Diet

The subjects had to undergo a 4-week weight loss intervention program, including reduced caloric intake and moderate to vigorous aerobic exercise. Each participant individually received detailed instructions and counselling about lifestyle changes, a personalised nutritional plan made using software PLANEAT (www.planeat.sk; accessed on 6 September 2014) and a plan of physical activity (Appendix A). Total daily calorie consumption consisted of 45% carbohydrates, 30% fat and 25% protein. In the WLPB group, their diet contained 30 g of the probiotic sheep cheese “Bryndza”. Altogether, 13 families, 24 genera and 44 species of microbiota were identified in Slovak cheese Bryndza [33]. The most abundant microorganisms in the Bryndza cheese are *Lactococcus*, *Streptococcus*, *Lactobacillus* and *Enterococcus* [21,26,34]. All the participants were asked to report their food consumption for feedback and control. Quantitative and qualitative data were analysed by the Planeat nutrition software (Planeat s.r.o, Bratislava, Slovakia). The food reports of all the participants were collected and analysed, and one random week was analysed and averaged.

#### 2.2.2. Physical Activity

There were structured and guided exercise training sessions three times a week throughout the intervention period. Each session, including the warm-up, strength training, cardiovascular training, cool down and stretching, lasted 30 min. Exercise was performed on hydraulic strengthening machines in the form of circuit training. One circuit consisted of 9 hydraulic machines and 9 exercise mats for aerobic activity. The load interval was set to 30 s of vigorous-intensity physical activity. One training session comprised 3 entire circuits and lasted approximately 30 min. In addition, the participants were advised to engage in moderate-intensity activity for at least 150 min per week [35].

### 2.3. Body Composition

Body composition characteristics were measured before and after the intervention. BMI was calculated as weight in kilograms divided by height in metres squared. We measured the body weight, height, body fat percentage, amount of fat mass and muscle mass using the bioelectrical impedance (Omron 511BF, OMRON HEALTHCARE Co., Ltd. Kyoto, Japan). Waist circumference was measured with a flexible tape.

### 2.4. Resting and Total Energy Expenditure

Resting metabolic rate (RMR) was measured using indirect calorimetry. Respiratory variables were continuously measured using a Cosmed K4b2 breath-by-breath gas analyser. Before measurement, the subjects were advised to get 7 h–8 h of sleep, to refrain from intense physical activity in the previous 24 h and to fast overnight before arriving for the examination. A flow meter calibration was conducted, and before each use the metabolic cart was calibrated with reference gas. After achieving a steady state (after 5 min), the expired gases were collected for 20 min and the final 10 min of the data were averaged; the resting metabolic rate was calculated using the Weir formula [36].

### 2.5. Stool and Blood Sample Analysis 

Faecal samples were collected from participants before and at the end of the intervention. Participants were instructed on how to prevent the contamination of samples during sample collection. They were provided with DNA/RNA Shield Fecal Collection Tubes for the collection and preservation of nucleic acids from stool specimens (ZymoResearch, Irvine, CA, USA). Samples were stored in the DNA/RNA Shield Fecal Collection Tubes at ambient temperature until delivered to the laboratory. Samples were aliquoted and stored at −80 °C.

### 2.6. DNA Extraction, High-Throughput Sequencing and Bioinformatics

Total DNA from the stool samples was extracted using the ZymoBiomics DNA/RNA mini kit (ZymoResearch Scientific, Irvine, CA, USA) in accordance with the manufacturer’s protocol. DNA from each sample was amplified using specific primers targeting the V1–V3 region of 16S rDNA [37]. PCR reaction contained 1 ng of DNA, 5xFIREPol MasterMix (Solis BioDyne, Tartu, Estonia) and 2  µM of each primer (10 µM). The reaction conditions for PCR amplification were 95 °C for 15 min; 25 cycles of 95 °C for 20 s, 56 °C for 30 s and 72 °C for 1 min; and final elongation at 72 °C for 5 min. The products of amplification were verified by agar electrophoresis. DNA libraries for Illumina sequencing were prepared using the index PCR reaction with input of 1 ng of DNA. The reaction conditions for the PCR were 95 °C for 15 min; 12 cycles of 95 °C for 10 s, 55 °C for 30 s and 72 °C for 90 s; and final elongation at 72 °C for 5 min. Index PCR amplification products were purified using 1.8x Agencourt AMPure XP magnetic beads (BeckmanCoulter, Brea, CA, USA). DNA libraries were validated by Agilent 2100 (Agilent Technologies, Santa Clara, CA, USA) and quantified by Qubit 2.0 Fluorometer (Thermo Fisher Scientific, Waltham, MA, USA). Mixed amplicons were pooled and sequenced using an Illumina MiSeq platform via a 300 bp paired-end reads Illumina sequencing system (Illumina, San Diego, CA, USA). 

### 2.7. Illumina Data Processing

Samples were analysed using the QIIME2 Core 2020.8 pipeline [38]. Quality control and feature table construction were performed using the DADA2 QIIME2 plugin [39] with default parameters except --p-trunc-len-r 280 and --p-max-ee-r 3. For taxonomic classification, we used a pre-trained naive Bayes classifier version 2020.6.1 [40] in the q2-feature-classifier QIIME2 plugin [41]. This classifier was trained on the Silva 132 99% OTUs full-length sequences [42].

### 2.8. Plasma Metabolite Concentrations 

The concentrations of the following 23 plasma metabolites were analysed: lactate, alanine, valine, leucine, isoleucine, glucose, creatinine, creatine, acetate, acetone, pyruvate, succinate, phenylalanine, tyrosine, glutamine, lysine, histidine, tryptophane, keto-leucine, keto-isoleucine, ketovaline and lipoprotein fractions Lipo1 (methyl [CH3] bands) and Lipo2 (long chain methylene [CH2] n bands) containing a fraction of VLDL, LDL, IDL and HDL. 

### 2.9. Blood Plasma Metabolites; NMR Data Acquisition

The plasma fraction was deproteinated by adding 600 µL of methanol to 300 µL of plasma. The mixture was vortexed for a few seconds and stored at −20 °C for 20 min. Subsequently, the mixture was centrifuged for 30 min at 14,000 rpm. Finally, 700 µL of supernatant was dried out and mixed with 100 µL of stock solution (150 mM phosphate buffer and 0.3 mM TSP-d4 3-(trimethylsilyl)-propionic-2,2,3,3-d4 acid sodium salt as a chemical shift reference in deuterated water) and 500 µL of deuterated water. The final mixture (550 µL) was transferred into a 5 mm NMR tube. 

NMR data were obtained by an Avance III 600 MHz NMR spectrometer equipped with cryoprobe (Brukker, Ettlingen, Germany). Initial settings were made from an independent sample and adopted for measurements. Before measurement, samples were stored in Sample Jet at 6 °C for no longer than 3 h and randomly ordered for acquisition. Measurements were carried out at 310 K. An exponential noise filter was used to introduce 0.3 Hz line broadening before the Fourier transform. We used standard Bruker profiling protocols with the following modifications: profiling: 1D NOESY with pre-saturation (noesygppr1d): FID size: 64 k, dummy scans: 4, number of scans: 128, spectral width: 20.4750 ppm; COSY with pre-saturation (cosygpprqf): FID size: 4k, dummy scans: 8, number of scans: 1, spectral width: 16.0125 ppm; homonuclear: J-resolved (jresgpprqf): FID size: 8 k, dummy scans: 16, number of scans: 4; profiling: CPMG with pre-saturation (cpmgpr1d, L4 = 126, d20 = 3 ms): FID size: 64k, dummy scans: 4, number of scans: 128, spectral width: 20.0156 ppm. All experiments were conducted with a relaxation delay of 4 s; all data were once zero filled.

### 2.10. Statistical Analysis

The data were explored and analysed by R ver. 4.0.3 [43], through the use of the emmeans [44], randomForestSRC [45] and pROC [46]. Exploratory data analysis involved data visualising by swarmplots overlaid with boxplots. Data were subjected to a two-way mixed ANOVA model followed by post hoc comparisons. A Random Forest (RF) machine learning algorithm was used to obtain the out-of-bag ROC curve for predicting pre/post status in the WLPB group and thus to estimate the discriminative ability of the selected physical, microbiome and metabolite markers. Machine learning techniques have several advantages over classical statistical models [47]. The correlations between the characteristics of the gut microbiota and metabolism were analysed using the Spearman correlation coefficient. The significance level of all statistical analyses was set at 0.05. Power calculations for this study were based on previous studies of effect of the probiotic intake on gut microbiota [48]. Assuming a mean of Bifidobacterium abundance 5.2(±4.6) in the placebo group and a standard mean of difference 4.6, our crossover design with *n* = 11 had a power of 0.80 at an alpha of 0.05 to detect at least 10% increase in the LB group following the diet with probiotics. 

ClustVis was used to visualise multidimensional data using principal component analysis (PCA) [49,50].

## 3. Results

### 3.1. Body Composition Analysis

Twenty-two participants completed the study. After completing the short-term weight loss program, significant differences were observed in the body composition characteristics in both groups (Table 1). There was significant decrease in body weight, BMI and body fat (%) in both WLPB and WLP groups. Furthermore, there was an increase in muscle mass (%) in both WLPB and WLP groups. There were no significant differences between weight, BMI, fat loss and muscle gain between the groups after intervention.

### 3.2. Nutrition Analysis

All the participants received a nutrition plan with food recipes for the duration of the intervention (Appendix A). The plan consisted of four meals per day. Based on the measured RMR, prescribed daily calorie consumption was 1526 ± 240 kcal, containing 45% carbohydrates, 30% fat and 25% protein. The food reports of all the participants were collected and analysed, and one random week was analysed and averaged. Total real calorie consumption and macronutrient intake was similar and did not differ between groups. It included 55–60% carbohydrates, 25–30% proteins and 18–20% lipids. Moreover, we did not find significant differences between the prescribed nutritional plan and self-reported food intake (Table 2).

### 3.3. Microbial Analysis

Gut microbiota composition was influenced after both interventions. Table 3 shows the significant differences in the abundance of certain microbial taxa after interventions. We identified 12 different phyla in the WLPB groups and 11 phylum taxa in the WLP groups. There were no significant changes in the relative abundance of bacteria at the phylum level within or between the groups. The microbiota alpha diversity of 18–65 year-old women defined by the Shannon, Simpson and Chao1 index did not differ within and between the groups.

The higher abundance of order *Erysipelotrichales* was measured within the groups (WLPB-pre vs. WLPB-post; *p* = 0.001; WLP-pre vs. WLP-post; *p* = 0.027). Moreover, family *Lachnospiraceae* decreased in the WLPB group (*p* = 0.006). It was also numerically decreased in the WLP group; however, the difference was not significant (*p* = 0.13). 

After an analysis of the relative abundance of LAB bacteria, we found some significant changes. In the WLPB group, we observed an increase in *Lactobacillales* and *Streptococcaceae.* Further at a lower taxonomic level, WLPB intervention increased the abundance of the genera *Lactococcus* and *Streptococcus* (Figure 1). However, no significant changes in the abundance of any LAB bacteria were observed after WLP intervention. We also found some bacterial shifts in SCFA-producing bacteria (Figure 1). *Phascolarctobacterium and Butyricimonas* increased at the genus level in the WLPB group compared to baseline (*p* = 0.019, *p* = 0.023 respectively).

Selected bacterial genera enabled the discrimination between the groups using principal component analysis (PCA) (Figure 2).

### 3.4. Plasma Metabolite Analysis

Appendix A shows the significant differences in the concentration of certain metabolites after the interventions. In the intra-group comparison, we identified a significant decrease in the lipoprotein fraction Lipo1 between WLPB-pre and WLPB-post (*p* = 0.036) and WLP-pre and WLP-post (*p* = 0.026). After the WLPB treatment there was a decrease in the Lipo2 fraction, although it did not reach significance (*p* = 0.055). Moreover, there was a significant decrease in isoleucine but only in the intra-group WLP. The decrease in isoleucine in the intra-group WLPB was not significant (*p* = 0.13). 

### 3.5. Machine Learning Analysis of Selected Variables/Predictors

Additionally, we have ranked selected physical, microbiome and metabolite markers used in the Random Forest (RF) machine learning (ML) analyses. The machine learning analysis identified *Lactococcus*, *Lactobacillales*, *Erysipelotrichales*, *Streptococcus*, Lipo1, BMI, *Lachnospiraceae* and muscle mass as the important predictors, and it led to an ROC curve with an AUC of 0.83, indicating a good ability to discriminate probands from WLPB-pre and WLPB-post groups (Figure 3). 

## 4. Discussion

We performed an intervention study to find the effect of a short-term weight loss program with vs. without Bryndza cheese consumption on the gut microbiota composition of healthy women. This is the first gut microbiome study to examine the combination of natural probiotics in the form of Bryndza and exercise training applied in a weight loss program. We hypothesised that a short-term weight loss program would have a positive influence on body composition and could favourably modulate gut microbiota. We expected that regular consumption of Bryndza cheese would have an additional effect on gut microbiota and metabolic parameters. Our main findings demonstrated similar effects of both intervention modalities, which led to decreased body weight, BMI index and body fat in both groups after completing a 4-week weight loss program. Furthermore, we found a higher abundance of the LAB bacteria after Bryndza consumption (*Lactobacillales*, *Streptococcaceae*, *Lactococcus* and *Streptococcus*). We also noticed an increase in SCFA-producing bacteria *Phascolacrtobacterium* and *Butyricimonas* after Bryndza consumption.

These data are in accordance with previous studies, where it was suggested that similar programs have a positive impact on the reduction of BMI and body fat [7,51,52]. Recent studies demonstrated that the intake of dairy products, such as milk and kefir, promoted higher body fat loss and lean mass gain in weight loss [53,54]. We recorded no significant differences between BMI, body fat and waist circumference loss between the WLPB and WLP groups since both recorded equal calorie consumption, which was confirmed by the calorie intake checks (Table 3). Interestingly, the positive influence of probiotics on weight loss is associated with decreased appetite due to the SCFA production of butyrate producing bacteria [55,56]. However, as mentioned above, the diet in our study was isocaloric for both groups. 

The main purpose of the study was to find out if a short-term weight loss program combined with Bryndza sheep cheese consumption would change the relative abundance of LAB. Šaková et al. used an independent method to identify some of the LAB bacteria in the Bryndza cheese that were mainly *Lactococcus* spp. followed by *Streptococcus* spp. and *Leuconostoc* spp. [22]. Based on this evidence, we expected a higher relative abundance of the LAB bacteria in stool samples after the probiotic intervention by Bryndza. We detected a significant increase in the relative abundance of the *Lactobacillales* population. Furthermore, the Random Forest machine learning analysis identified *Lactococcus* as the most important predictor in a small group of predictors with a good ability to discriminate between the subjects from WLPB-pre and WLPB-post groups. This taxa is associated with improved gut integrity, reduced gut permeability and induced anti-inflammatory response [57,58]. In another study, *Lactobacillales* were increased after kefir administration in patients with metabolic syndrome [59]. After a short-term weight loss program with Bryndza cheese consumption, we found a significant increase in additional LAB bacteria, specifically *Lactococcus* and *Streptococcus*. These genera are commonly used as starter cultures in fermented dairy foods [60]. This is in accordance with previous studies reporting an increased abundance of *Lactococcus* and *Streptococcus* as a result of the high consumption of fermented milk products [30,61]. *Lactococcus* is a homolactic fermentative bacteria that converts its carbon source to lactate from pyruvate with lactate dehydrogenase enzyme utilisation [62]. This genus does not gain too much interest for its probiotic activity because the main attention is focused on the other LAB genera as *Lactobacillus* and *Bifidobacterium* [63]. However, some studies have shown the beneficial effects of products fermented with *Lactococcus* in the gut [64]. A previous study on subjects with slightly elevated blood lipid or blood sugar levels demonstrated that the administration of yogurt fermented with the *Lactococcus lactis* 11/19-B1 strain for 8 weeks significantly reduced total cholesterol, LDL and LDL/HDL ratio [64]. Moreover, 8 weeks of drinking skim milk fermented with *Lactococcus* genera has shown a positive impact on the systolic blood pressure level in prehypertensive individuals [65]. There is also some evidence to suggest that the consumption of fermented dairy foods influences the abundance of *Streptococcus* genera in the gut. Burton et al. showed a significant increase in *Streptococcus* after a 2-week intervention with probiotic yogurt in young males [66]. Similar to *Lactococcus*, it has been reported that a higher abundance of *Streptococcus,* induced by fermented milk intake, improved the lipid profile in cholesterolaemic adults [67]. These results agree with our findings of increased *Lactococcus* population in the WLPB group. Although our participants were non-obese, we found a significant decrease in lipoprotein fraction 1 after the weight loss program in both WLPB and WLP groups. However, we also found decreased lipoprotein fraction 2 that almost reached statistical significance (*p* = 0.055) in the WLPB group. In previous studies it was demonstrated that probiotic bacteria had a favourable impact on lipid profile [68,69,70]. In this regard, we believe that Bryndza consumption could represent an added benefit in reducing the lipoprotein fraction.

Furthermore, we detected a decrease in the relative abundance of metabolite isoleucine in the WLP intra-group. We suggest that it could be attributed to the negative energy balance of the short-term weight loss program. These results are in agreement with a previous study on Japanese adults with metabolic syndrome, where they observed the significant correlation between weight loss and reduction of isoleucine concentration [71]. However, in the WLPB group, who consumed the Bryndza cheese, the decrease in isoleucine did not reach a significant value (*p* = 0.13). Amino acid isoleucine is abundant in whey proteins and ewe’s milk [72]. Rigamonti et al. demonstrated that the administration of whey protein significantly increased the concentration of isoleucine and other amino acids in young, obese women [73]. Therefore, we believe that the protein content in Bryndza may compensate for the catabolic effect due to the weight loss program.

Fermented dairy foods affect the abundance of bacteria producing short chain fatty acids (SCFA) [74]. In our study, *Butyricimonas* and *Phascolarctobacterium,* SCFA producers, were elevated in the WLPB intra-group. A recent clinical study revealed a higher abundance of *Phascolarctobacterium* after the intervention of Kimchi, which is another lactic acid bacteria-fermented food [75]. Previous studies demonstrated the positive correlation between these taxa and an amelioration of the glucose homeostasis [76,77]. Moreno-indias et al. demonstrated a negative correlation between *Butyricimonas* and plasma glucose. They also found an elevated level of *Phascolarctobacterium* in appendix samples with insulin sensitivity compared to insulin-resistant obese individuals [77]. These data agree with our results, where we found a negative correlation between the relative abundance of *Butyricimonas* and fasting insulin level after the WLPB intervention. Moreover, Naderpoor et al. [76] showed a positive correlation between the higher abundance of *Phascolarctobacterium* and insulin sensitivity in overweight adults [76]. We did not find any significant differences in fasting glucose levels, probably due to the normal glucose levels of the subjects. 

We assumed that the short-term weight loss program without Bryndza sheep cheese would, to some extent, affect the composition of gut microbiota too, as there is evidence to suggest that weight loss programs may affect the composition of gut microbiota in humans [78,79,80]. Our microbial analysis did not reveal many significant bacterial shifts after the short-term weight loss program. This may be due to the fact that the participants in our study were not obese and lost an average 3.65 ± 1.70 kg (WLPB) and 3.66 ± 2.50 kg (WLP) of body weight. However, the interesting finding is that the population of *Erysipelotrichales* increased significantly in both WLPB and WLP groups after intervention, which may be explained by a response to the weight loss program rather than the probiotic intervention. Interestingly the Random Forest machine learning analysis identified *Erysipelotrichales* as the third most important discrimination factor for subjects from WLPB-pre and WLPB-post groups. The increased number of *Erysipelotrichaceae* was previously measured in healthy subjects with higher cardiorespiratory fitness levels [81]. Even though we did not examine the physical fitness of our responders, we suppose that 4 weeks of exercise training might increase the physical fitness in both groups. Furthermore, the *Erysipelotrichaceae* family is considered the key SCFA producer [81]. The gut bacteria-derived SCFA play a role in maintaining intestinal and immune homeostasis and anti-inflammatory response [82,83]. 

We also identified a decrease in the *Lachnospiraceae* family in the WLPB and WLP intra-groups. There is strong evidence to suggest that *Lachnospiraceae* were positively associated with certain metabolic disturbances [84,85,86,87]. In an observational study, Chávez-Carbajal et al. confirmed that *Lachnospiraceae* was significantly more abundant in obese and extremely obese women [85]. In addition, *Lachnospiraceae* count is positively associated with diet-related obesity in British men [86]. These results are consistent with our data, where we report decreased weight, body fat and BMI. 

Our findings substantiate the benefits of a physically active lifestyle and diet. We strongly recommend further research into the combination of other natural probiotics and physical activity for health promotion and disease prevention.

The main limitation of our study is the small number of participants. We observed elevated drop out from the study mainly due to seasonal flu or non-compliance. However, the majority of participants who completed the study were in the Bryndza group. Further long-term clinical trials with the diet containing a natural probiotic as a weight loss intervention are necessary to investigate the promising protective effect of this intervention on serum/plasma metabolites, body composition and health. The second limitation of our study was age variance. However, all of the subjects were highly motivated to improve their diet and lose body weight. Third, we did not measure the exact amount of physical activity during the intervention.

## 5. Conclusions

Short-term weight loss intervention both with and without Bryndza cheese consumption improves body composition and increases the count of SCFA producers, although Bryndza consumption results further the health benefits by increasing the relative abundance of lactic acid bacteria. From the broader perspective, our findings suggest health benefits of consuming natural probiotics, such as Bryndza cheese, and emphasise their role in the diet of modern Man.

## Figures and Tables

**Figure 1 nutrients-13-01753-f001:**
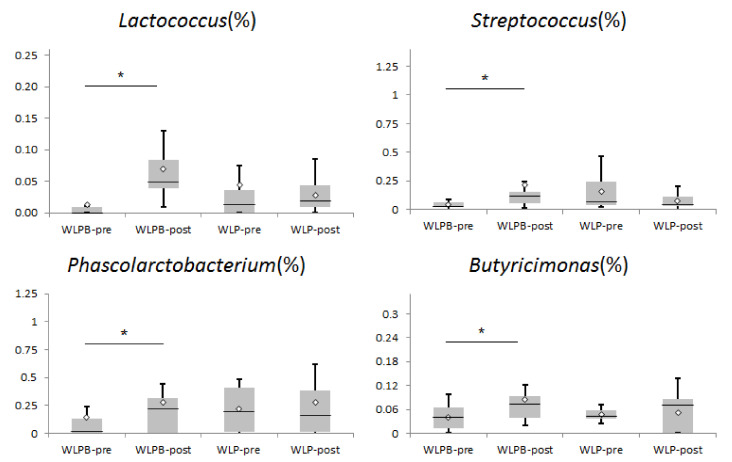
Relative abundance of gut bacteria taxa in the WLP and WLPB groups compared to baseline. WLPB—short-term weight loss program + Bryndza consumption; WLP—short-term weight loss program. * *p* < 0.05.

**Figure 2 nutrients-13-01753-f002:**
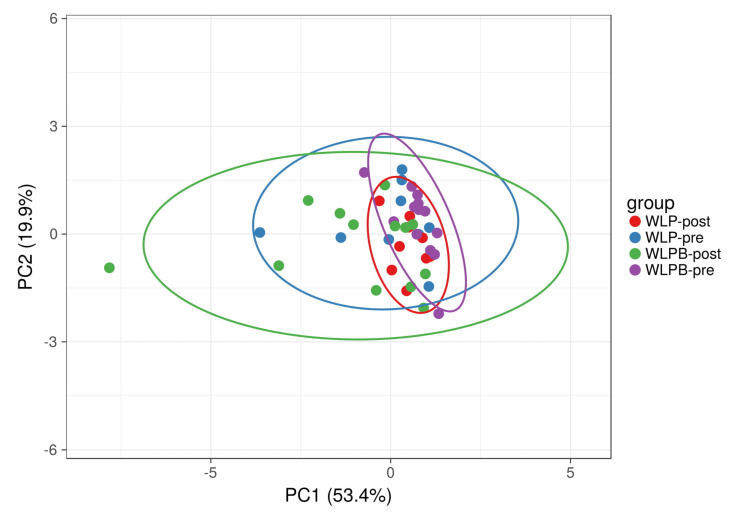
Beta diversity of analysed samples represented by significantly altered (*p* < 0.05) bacterial taxa, selected by a Random Forest machine learning analysis, before and after a short-term weight loss program in Bryndza and control groups visualised by PCA. SVD with imputation is used to calculate principal components. The X and Y axis show principal component 1 and principal component 2 that explain 53.4% and 19.9% of the total variance, respectively. Prediction ellipses are such that with a probability of 0.95, a new observation from the same group will fall inside the ellipse (*n* = 44 data points).

**Figure 3 nutrients-13-01753-f003:**
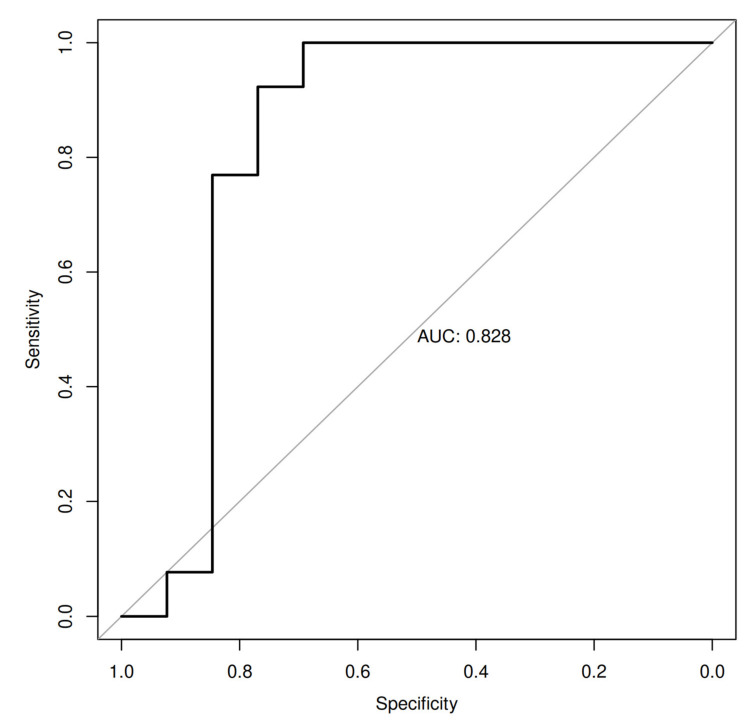
ROC (receiver operating characteristic) curves with the area under the ROC curve (AUC) for the RF-ML algorithm with Lactococcus, Lactobacillales, Erysipelotrichales, Streptococcus, Lipo1, BMI, Lachnospiraceae and muscle mass identified as the important predictors for discriminating between WLPB-pre and WLPB-post groups. Abbreviations: FPR, false positive rate; RFM-L, Random Forest machine learning; TPR, true positive rate.

**Table 1 nutrients-13-01753-t001:** Physical characteristics at baseline and after completing the short-term weight loss program.

	**WLPB-Pre (*n* = 13)**	**WLPB-Post (*n* = 13)**	***p*** **-Value**
Weight (kg)	71.9 (±15.1)	68.2 (±14.0)	0.001
BMI(kg/m^2^)	26.0 (±5.6)	24.5 (±5.4)	0.003
Waist (cm)	82.9 (±12.4)	77.8 (±9.2)	0.004
Body fat (%)	33.9 (±8.8)	31.6 (±8.9)	0.003
Muscle (%)	28.9 (±3.9)	29.8 (±4.1)	0.016
	**WLP-Pre (*n* = 9)**	**WLP-Post (*n* = 9)**	***p*** **-Value**
Weight (kg)	71.9 (±14.2)	68.2 (±13.4)	0.002
BMI(kg/m^2^)	24.8 (±4.9)	23.5 (±4.5)	0.003
Waist (cm)	76.4 (±6.4)	70.3 (±5.6)	0.026
Body fat (%)	31.0 (±4.2)	26.9 (±2.9)	0.012
Muscle (%)	29.3 (±1.1)	31.4 (±0.4)	0.043

Values are presented as mean (±stdev). WLPB—short-term weight loss program + Bryndza consumption; WLP—short-term weight loss program.

**Table 2 nutrients-13-01753-t002:** Calorie and nutrient intake calculated from self-reported food intake.

	PNP	WLPB (*n* = 13)	WLP (*n* = 9)
Energy (kcal·kg^−1^)	21.2 (±0.4)	21.1 (±3.5)	20.3 (±3.2)
Carbs (g·kg^−1^)	2.3 (±0.4)	2.3 (±0.4)	2.2 (±0.4)
Protein (g·kg^−1^)	1.3 (±0.2)	1.1 (±0.2)	1.1 (±0.2)
Lipids (g·kg^−1^)	0.7 (±0.1)	0.8 (±0.1)	0.7 (±0.1)
RMR (kcal·kg^−1^)		21.8 (±4.8)	21.2 (±1.5)

Values are presented as mean (±stdev). PNP—prescribed nutritional plan; WLPB—short-term weight loss program + Bryndza consumption; WLP—short-term weight loss program.

**Table 3 nutrients-13-01753-t003:** Microbial taxa (order, family, genus) differentially present after interventions.

**Taxon (%)**	**WLPB-Pre (*n* = 13)**	**WLPB-Post (*n* = 13)**	***p*-Value**
*Lachnospirales*	41.4751(±11.691)	35.7855(±11.771)	0.006
*Bacilli*	0.9575 (±1.3003)	2.3970 (±2.0402)	0.006
*Erysipelotrichales*	0.7941 (±1.1511)	1.8820(±1.8926)	0.006
*Lactobacillales*	0.0731 (±0.0582)	0.3101 (0.3302)	0.001
*Lachnospiraceae*	0.4063 (±0.4462)	0.4096(0.5498)	0.006
*Erysipelatoclostridiaceae*	0.2443 (±0.2682)	0.8713 (±1.2429)	0.006
*Erysipelotrichaceae*	0.5497 (±1.1691)	1.0107 (±1.4469)	0.019
*Streptococcaceae*	0.0590 (±0.0540)	0.2890 (±0.3228)	0.001
*Lactococcus*	0.0138 (±0.0313)	0.0705 (±0.0537)	0.013
*Streptococcus*	0.0451 (±0.0378)	0.2184 (±0.3281)	0.011
*Butyricimonas*	0.0401 (±0.0333)	0.0854 (±0.0781)	0.023
*Phascolarctobacterium*	0.1430 (±0.2786)	0.2819 (±0.3530)	0.019
*Alistipes*	1.0346 (±0.9398)	2.3725 (±2.3569)	0.013
*Turicibacter*	0.1159 (±0.2062)	0.2672 (±0.3028)	0.034
*Erysipelotrichaceae UCG-003*	0.2021 (±0.2710)	0.8086(±1.2434)	0.002
*Prevotella*	0.5499 (±0.8897)	0.1679 (±0.2023)	0.033
*Lachnospiraceae UCG-004*	0.2109 (±0.0958)	0.3175 (±0.050)	0.002
*Roseburia*	4.0680 (±1.8758)	2.6113 (±1.1804)	0.006
**Taxon (%)**	**WLP-Pre (*n* = 9)**	**WLP-Post (*n* = 9)**	***p*-Value**
*Erysipelotrichales*	0.5232 (±0.2133)	0.9384 (±0.5941)	0.027
*Peptostreptococcaceae*	0.1841 (±0.0849)	0.0792 (±0.0564)	0.014
*Lachnospiraceae UCG-010*	0.0173 (±0.0086)	0.0086 (±0.0076)	0.017
*Marvinbryantia*	0.2259 (±0.1041)	0.1056 (±0.0878)	0.035
*Lachnospiraceae NK3A20group*	0.0328 (±0.0257)	0.0118 (±0.0168)	0.015

Values are presented as mean (±stdev). WLPB—short-term weight loss program + May Bryndza consumption; WLP—short-term weight loss program.

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
