# Peer review of "The Efficacy of Short-Term Weight Loss Programs and Consumption of Natural Probiotic Bryndza Cheese on Gut Microbiota Composition in Women"

_nutrients, 2021, doi:10.3390/nu13061753_

Round 1
Reviewer 1 Report
In the manuscript nutrients-1222089 titled “The Efficacy of Short-Term Weight Loss Programs and Consumption of Natural Probiotic Bryndza Cheese on Gut Microbiota Composition in Women” by Ivan Hric and colleagues, they have reported that the short-term weight loss program with Bryndza cheese consumption improves body composition and increases the abundance of lactic acid bacteria and short-chain fatty acids producers. I have few comments regarding the present manuscript
-The manuscript seems a very organized work, the most relevant is the detailed information in the materials and methods section, I read with interest the microbial analysis and the authors have stated that they used DADA2, and in the last paragraph, they talk about OTU, using DADA2 pipeline they have obtained ASV instead OTU, please check the information.
-How the authors have calculated the samples size in the present study
-Principal coordinate analysis using the microbiota analysis is required in this manuscript
-Recently I read an interesting analysis to obtain the ROC curve with the microbiota data, the analysis is available in the R package as selbal and the name of the analysis is Rivera Pinto, maybe the authors could take advantage of the aforementioned analysis
-How the diet or even physical exercise impacts the reported data? please explain these important relationships
Author Response
-The manuscript seems a very organized work, the most relevant is the detailed information in the materials and methods section, I read with interest the microbial analysis and the authors have stated that they used DADA2, and in the last paragraph, they talk about OTU, using DADA2 pipeline they have obtained ASV instead OTU, please check the information.
A: Thanks for the detailed review. We believe that the corrections will contribute to greater clarity and interest of readers. The term OTUs is in this context appropriate since it does not mean the taxonomical unit, but is part of the name of the Silva database version used for Naive Bayes classifier training.
-How the authors have calculated the samples size in the present study
A: Power calculations for this study were based on previous study of effect of the probiotics intake on gut microbiota referred in manuscript. We calculate sample size using https://clincalc.com/stats/samplesize.aspx.
-Principal coordinate analysis using the microbiota analysis is required in this manuscript
A: We agree and added visualization in revised manuscript. Since the measured variables are continuous, we have performed euclidean PCoA, which is the same as PCA.
-Recently I read an interesting analysis to obtain the ROC curve with the microbiota data, the analysis is available in the R package as selbal and the name of the analysis is Rivera Pinto, maybe the authors could take advantage of the aforementioned analysis
A: Thank you very much for pointing us to selbal library and the paper. Since it looks very interesting, we plan to explore the approach and utilize it in our future analyses. We provide a reference to the paper and R library in revised manuscript (Statistical analysis).
-How the diet or even physical exercise impacts the reported data? Please explain these important relationships
A: Well, thank you for this interesting question. Relying on the results of a control group where we applied the same diet and exercise as in experimental group we may conclude that 4 weeks is short period for extensive microbiota changes. Based on our experience with data collection, we admit that it has been easier to hire and motivate women to experiment when a complex weight loss program (exercise and diet) is available. Anyway to answer the reviewer question we suppose that short-term diet or even physical exercise impacts gut microbiome in very limited way.
Reviewer 2 Report
Manuscript nutrients-1222089, entitled “The Efficacy of Short-Term Weight Loss Programs and Consumption of Natural Probiotic Bryndza Cheese on Gut Microbiota Composition in Women”
General comment
The article provides useful information about the efficacy of short-term weight loss programs and consumption of natural probiotic Bryndza cheese on gut microbiota composition in women. Although the experiment is in general appropriately designed and implemented, there are some points that should be corrected or clarified.
Major comments
- What are the effects of Bryndza Cheese consumption without a short-term weight loss program?
- Bryndza or May Bryndza Cheese?
Minor points:
L29-30: “…lead to significant and favourable change of BMI, body fat…”
L34: “…in WLPB. A short-term weight loss program combined with Bryndza…”
L36: “…producers in middle-aged women.”
L52-54: Please rephrase
L62: “…in the defined…”
L66-68: References 26-28 are not all from Pangallo et al.
L74: “At the same time, we aimed at elucidating benefits…”
L80: “allocated” instead of “divided”
L96: “received” instead of “get”
L98: Please delete “which”
L100: Please delete “the food”
L103, 227: “All the participants…”
L106, 231: “…were collected and analysed.”
L109: “There were structured and guided exercise training sessions…”
L129: “advised” instead of “told”
L196: “…analysed by R…”
L218-220: Please delete
L250-251: “…Lachnospiraceae was decreased in WLPB (p=0.006). It was also numerically decreased in the WLP group, however, the difference was not significant (p=0.13).”
L268-273: Please remove to Materials and Methods
L303: “Our main findings demonstrated similar effects…”
L304: “lead” instead of “leads”
L319-320: “…weight loss program combined with Bryndza sheep…”
L322: “…culture that were mainly Lactococcus…”
L334: “…reporting an…”
L338: “does not gain” instead of “hasn’t got”
L360: “We suggest that it could be attributed to the…”
L366: “Rigamonti et al. demonstrated…”
L371: “…affect the abundance…”
L372: “In our study, Butyricimonas…”
L382: “…showed a positive…”
L383: Please delete “[74]”
L401: “Furthermore” instead of “Further”
L402: “producer” instead of “producent”
L407: “Chávez-Carbajal et al. confirmed…”
L408: “In addition, Lachnospiraceae count is positively associated…”
L426: “results in” instead of “brings”
Author Response
Major comments
- What are the effects of Bryndza Cheese consumption without a short-term weight loss program?
A: Well, thank you for this interesting question. Based on our experience with data collection, we admit that it has been easier to hire and motivate women to experiment when a weight loss program is available. Being motivated with the results of this study, we are completing a bryndza experiment with young swimmers, in which the consumption of bryndza is the “only” factor distinguishing experimental and control group. Anyway to answer the reviewer question, based on probiotic studies, we can assume an increase in lactic acid bacteria and butyrate producers.
Bryndza or May Bryndza Cheese?
A: May bryndza cheese is a highly prized variant of bryndza, which is produced at the beginning of the summer season in May. It is assumed that the production period has a positive effect on the quality of the cheese, probably the quality of sheep's milk affected by spring pastures. We completed the experiment in May, so we used the term May Bryndza Cheese which was in accordance with previously published papers. To avoid possible misunderstandings, we use the name Bryndza cheese throughout the revised manuscript.
Minor points:
Thank you for your detailed handwriting review. All minor points have been corrected or reworked.
Round 2
Reviewer 1 Report
Thank you to the authors for taking into account my previous comments, the manuscript now reads well and the questions were well described